# Antibiotic Stewardship in Retail Pharmacies and the Access-Excess Challenge in China: A Policy Review

**DOI:** 10.3390/antibiotics11020141

**Published:** 2022-01-21

**Authors:** Tingting Zhang, Helen Lambert, Linhai Zhao, Rong Liu, Xingrong Shen, Debin Wang, Christie Cabral

**Affiliations:** 1Population Health Sciences, Bristol Medical School, University of Bristol, Bristol BS8 2PN, UK; h.lambert@bristol.ac.uk (H.L.); christie.cabral@bristol.ac.uk (C.C.); 2School of Health Services Management, Anhui Medical University, Hefei 230032, China; zhaolinhai@ahmu.edu.cn (L.Z.); liurong@ahmu.edu.cn (R.L.); shenxr@ahmu.edu.cn (X.S.); wangdebin@ahmu.edu.cn (D.W.)

**Keywords:** antimicrobial resistance, antibiotic, pharmacy, China, policy

## Abstract

In China, efforts to restrict excessive antibiotic consumption may prevent sufficient access to these life-saving drugs among the most deprived in society because of the weak primary health care system. This makes antibiotic stewardship in the retail pharmacy sector a particular challenge. We conducted an analysis to examinate policies on antibiotic sales in retail pharmacies in China and how tensions between ‘excess’ and ‘access’ are managed. The analysis was guided by the Walt and Gilson health policy analysis triangle to systematically analyse policies based on the content of policies, contexts, governance processes, and actors. Nine research studies and 25 documents identified from national and international sources were extracted, grouped into categories, and examined within and across records and categories. As of 2020, eight key policies have been introduced in China that focus on two areas: dispending prescribed medicines or antimicrobials with a prescription and having a licensed pharmacist present in the retail pharmacies, with approaches having changed over time. Inappropriate sales of antibiotics are still common in retail pharmacies, which can be linked to the lack of consistency and enforcement of published policies, the profit-driven nature of retail pharmacies, and the displacement of the demand for antibiotics from clinical into less regulated settings.

## 1. Introduction

Pathogenic resistance to antimicrobials and in particular, antibiotic resistance, is a serious public health threat and causes a major economic burden worldwide [1,2]. This problem has developed mainly as a result of the overuse of antibiotics in humans, agriculture, and the environment [3]. Low and middle-income countries (LMICs) including China have a high level of overuse of antibiotics for human health and, in turn, many common bacteria show high resistance rates to antibiotics [4,5,6]. However, access to quality primary health care and essential medicines is limited in many LMIC contexts and the burden of communicable diseases is high, so there is a need to balance access to life saving antibiotics against excess use [7,8]. There is a risk that efforts to control access to antibiotics to restrict excessive consumption may prevent the poor from having sufficient access to these life-saving drugs, which makes regulating antibiotic use a particular challenge in China and other LMICs.

In many LMICs, there are strong private pharmaceutical sectors to compensate for weak health systems; patients commonly purchase antibiotics without a prescription as the first step in seeking health care for an infection [9,10]. LMICs have issued policies to restrict over the counter (OTC) sales of antimicrobials in private pharmacies, but the impact is varied. These have been found to be effective in significantly decreasing the national usage of antimicrobials and/or changing their consumption trend in some Latin American countries [11,12,13,14]. However, these policies are not enacted in other LMICs such as India where, despite being illegal, customers purchase antimicrobials without a prescription from private pharmacies across the country [15]. 

In China, the retail pharmacy sector has grown rapidly over the last few decades and has become the main source of OTC antibiotics, both improving access and contributing to the overuse of antibiotics [16]. The cost of antibiotics can be covered by health insurance. China’s health insurance system is based around three basic medical insurance schemes and covers over 95% of the population including Urban Employee Basic Medical Insurance (UEBMI) and Urban Resident Basic Medical Insurance (URBMI) for urban areas and the New Rural Cooperative Medical Scheme (NRCMS) for rural areas. Patients covered by UEBMI can use their medical savings account to pay for outpatient medical services and medications, while URBMI and NRCMS will cover approximately 50% (varied in regions) of outpatient medical services and medications for patients who are members of these schemes [17]. Although antibiotic stewardship in China began in the 2000s, policies regulating the use of antibiotics were enforced from 2011 and have predominantly focused on hospitals and primary health care, whereas policies regulating retail sales of antibiotics are less frequently issued and weakly enforced [18]. There is limited research that has examined the policies aiming to regulate antibiotics in retail contexts. The aim of this paper was to undertake a policy analysis of the regulations and strategies that address antibiotic sales in retail pharmacies in the complex context of China and are relevant to the issue of excess-access.

## 2. Results

A total of nine research studies and 25 documents including regulations, strategies, reports, and guidelines were included in this policy review (Appendix A). As of 2020, in total eight key policies have been introduced: four of them are regulations that were issued by the China Food and Drug Administration (CFDA), the other four are strategies issued by CFDA, State Council, and National Health and Family Planning Commission to provide guidance on the enforcement of regulations. These policies were introduced and implemented over the last two decades as part of the efforts made by the national government to regulate antibiotic use in China.

### 2.1. Context

A weak primary health care system and strong retail pharmacy sector contribute to the difficulties of regulating access to antibiotics. In China, primary health care nearly broke down when the economic reforms of the late 1970s led to the collapse of people’s communes and state-owned enterprises, which were the foundations of China’s old health care system, and dramatically reduced the national subsidies for health care [17,19]. The national government rebooted primary health care in the 2000s, but this system is relatively weak, particularly in poorer rural areas of China [20]. Additionally, in the late 1970s, the Chinese pharmaceutical market started to grow rapidly due to the loss of direct central control and an ‘open door’ policy introduced in economic reforms [21,22]. Pharmaceutical sales by retail pharmacies grew by an average of 20% each year from 1978 to 2009 [23], and, by 2016, there were over 450,000 retail pharmacies in China [24]. The total annual revenues of retail pharmacies were estimated to reach over CNY 1 trillion (US $157 billion) by 2020 [17]. Retail pharmacies became one of the main sources of obtaining antibiotics, and antibiotics were found to have the highest average sales volume among all medicines in some retail pharmacies [16,25].

The development of national-level policies related to antibiotic stewardship in China began in the 2000s, but lacked legal power and were weakly implemented [26,27,28]. This changed in 2009 when China began a new round of health care reforms and introduced five pillars to achieve universal health coverage to all residents: strengthen primary health care, increase health care insurance coverage, ensure access to essential medicines, promote public health, and reform public hospitals [29]. The reforms contributed to policies promoting progress towards the rational use of antibiotics [30]. Coupled with these reforms, a new strategy, the National Special Campaign, ran from 2011 to 2013 and centred on the regulations for the clinical use of antibiotics. These policies focused on the clinical setting with clear tasks and responsibilities related to the appropriate use of antibiotics set up for physicians, pharmacists, microbiologists, and administrators in health care institutions [31,32,33,34]. Following this 3-year period, an improvement in hospital antibiotic use was observed [35,36]. In 2016, a more comprehensive five-year national action plan was issued to curb antibiotic resistance through the rational use of antimicrobials in multiple areas [37]. 

### 2.2. Policy Actors

Prior to the economic reforms, most pharmaceutical enterprises were state-owned. The state government established a central department, the Pharmaceutical Administration (PA), in 1978 to directly control the entire pharmaceutical industry in China. However, the economic reforms encouraged private capital and competition, which tended to give decision making power to the managers of pharmaceutical enterprises while PA became a less powerful regulating authority. Furthermore, PA’s principal responsibility is to support pharmaceutical enterprise development; it was also allowed to participate in profit-making activities and received financial support from the regulated pharmaceutical enterprises [21]. This affected the PA motivation to regulate pharmaceutical activities and over time, pharmaceutical enterprises became independent business entities. 

In 1998, a new administrative authority, the China Drug Administration (CDA), was established by grouping PA together with other two pharmaceutical related authorities. It was then replaced by the CFDA in 2003. CFDA was directly governed by the State Council (2003–2008), then the Ministry of Health (2008–2013), and then turned back to be governed by the State Council again after 2013; its main responsibility was to publish and implement regulations and to guide and monitor food and medicine related practices including all medicine related activities in retail pharmacies. CFDA was not allowed to make any profits from pharmaceutical entities [38]. In 2018, this authority was replaced by the State Administration for Market Regulation and National Medical Products Administration, which are also independent non-profit public sector regulatory authorities.

### 2.3. Content and Process

Policies of antibiotic sales mainly focused on two domains: dispending prescribed medicines or antimicrobials with a prescription and having a licensed pharmacist present in retail pharmacies. A timeline of these key policies is shown in Figure 1, with a brief summary of each policy given in Table A1. Most policies are about prescribed medicine administration and management in general, which includes antimicrobials, with only two policies focusing specifically on antimicrobials introduced in 2003 and 2016.

The current requirements are that (since 2015) all retail pharmacies must have a licensed pharmacist on staff and present during business hours, and that (since 2020) all antimicrobials must be dispensed with a prescription in all retail pharmacies.

#### 2.3.1. Dispensing Prescribed Medicines or Antimicrobials with a Prescription

During the 2000s, the national government issued a series of regulations aimed at controlling the management of prescribed medicines generally. These regulations make no specific mention of antimicrobials. The list of non-prescribed medicines that can be sold OTC was published separately by CFDA and does not include any oral antimicrobials; only a few external use antimicrobials such as Erythromycin Eye Ointment were included in Class A non-prescribed medicine [39]. Therefore, these regulations for prescribed medicines apply to all oral antibiotics.

The regulation, ‘*Measures for Separate Administration of Prescribed and Non-prescribed Medicines (Elementary edition)*’, implemented on 1 January 2000, is the first government policy to set out different procedures for prescribed and non-prescribed medicines. This regulation stated that prescribed medicines can only be obtained with a prescription from a licensed physician or licensed physician assistant. People who have a medical related education background and experience and pass the relevant national examinations can be licensed to work in health care facilities. Among them, those who complete university or higher-level qualifications can become a licensed physician, while those with college or lower levels can become a licensed physician assistant. The CFDA is responsible for classifying and issuing the list of non-prescribed medicines that are approved for OTC sale, and in cooperation with local Food and Drug Administrations (FDAs) to inspect and ensure the implementation of regulations [40]. On the same date, the linked ‘*Temporary Provisions for Prescribed and Non-prescribed Medicine Distribution*’ was implemented. In this regulation, more specific requirements of retail pharmacies were provided on how to procure, display, dispense, and record prescribed medicines; it required retail pharmacies to display prescribed medicines separately and for qualified staff to check and sign prescriptions before any can be dispensed [41]. The regulation ‘*Measures for the Supervision and Administration of Circulation of Pharmaceuticals (Elementary edition)*’ was introduced in 1999 and set out the legal consequences of violating relevant requirements for pharmaceutical manufacturers, wholesalers, retail pharmacies, and health care facilities. However, it did not state the legal penalties for the particular action of not managing prescribed medicines as per the regulations required in the retail pharmacies until its final edition was implemented in 2007 [42,43]. Since 2007, dispensing prescribed medicines without a prescription or in the absence of a licensed pharmacist/pharmaceutical technician will result in a warning; a further financial penalty of CNY 1000 (US $157) can be applied for retail pharmacies who seriously violate or do not correct their practices [43]. 

The strategy ‘*2004–2005 Work Plan related to Separate Administration of Prescribed and Non-prescribed Medicines*’ published in 2004 summarises the achievement since relevant policies were issued from 1999: the initial establishment of the pharmaceutical system that separates the management of prescribed medicines from non-prescribed medicines. This strategy also set a goal that, by the end of 2005, all prescribed medicines will be purchased only with a prescription and dispensed and used under the guidance of a pharmacist. For retail pharmacies that failed to manage prescribed medicines correctly, their business scope would be narrowed to non-prescribed medicines only. For rural areas with difficulties in following these regulations due to poverty and a lack of appropriately qualified pharmaceutical staff, however, this work plan stated that local authorities may develop their own prescribed medicine list to apply to local retail pharmacies, which would enable the public to access a larger range of prescribed medicines in a more convenient way. This national policy also set up a series of action plans for other relevant stakeholders in 2004–2005 including completing the legal and regulatory system to administer prescription only medicines and increasing public awareness of the harm of irrational use and overuse of medicines [44].

Although these policies to regulate prescribed medicines have applied to antimicrobials and, theoretically at least, retail pharmacies have not been allowed to sell oral antibiotics without prescription from 2000, the national government introduced two additional policies focusing specifically on the sale of antimicrobials in retail pharmacies, one in 2003 and one in 2016, indicating the difficulty of regulation in this context. In 2003, a national policy, ‘*Notice on Strengthening Administration over the Sale of Antimicrobials in Retail Pharmacies to Promote Rational Use of Medicines*’, stated that from 1 July 2004, all antimicrobials not on the non-prescribed medicines list (i.e., all oral antibiotics) could only be dispensed with a licensed physician’s prescription. This policy also clarified the roles of FDAs to ensure implementation of the policy by notifying retail pharmacies and inspecting their work, and to educate the public about the rational use of antimicrobials under the guidance of health professionals [45]. After a set of policies issued since 2011 to regulate the use of antibiotics in hospitals, the most comprehensive national plan to tackle antimicrobial resistance, ‘*The National Action Plan to Curb Bacterial Resistance (2016–2020)*’, was announced in 2016. In this policy, the sale of antimicrobials in retail pharmacies was emphasised as an important focus for curbing antimicrobial overuse. The government set a goal that by 2020, 100% of retail pharmacies should dispense antimicrobials by prescription only, and has highlighted strict enforcement of the regulations. The plan also stated that the severity of punishments for violations would be increased; however, details of these punishments have not been issued [37].

#### 2.3.2. Licensed Pharmacists in Retail Pharmacies

In China, licensed pharmacists are covered by the professional qualification system, which is overseen by the CFDA and Ministry of Human Resources and Social Security [46]. A pharmaceutical technician is required to have a high school diploma or equivalent and 3–6 months of pharmacy related training at the college level [23]. To become a licensed pharmacist working in a retail pharmacy, the pharmaceutical technician needs to pass the national exam to obtain a Licensed Pharmacist Certificate and register their certificate with a certain retail pharmacy through the provincial regulatory authorities [46]. Re-registration is required every three years and licensed pharmacists must complete continuous professional development training to gain approval for re-registration [46,47]. 

According to the regulations, since 2000, retail pharmacies should have stopped dispensing prescribed medicines if they do not have qualified pharmacy staff or if these staff are absent during business hours [41,42,43]. In 2004, the regulation ‘*Measures for the Administration of Pharmaceutical Trade License*’ further stated that the employment and presence of licensed pharmacists or pharmaceutical technicians is a pre-requisite for applying for a Pharmaceutical Trade License, which is an essential certificate to operate a retail pharmacy and sell prescribed medicines [48,49]. However, the government recognised the difficulties of requiring all retail pharmacies to align with these new regulations immediately, partly because of the insufficient and unequal distribution of pharmacists in China, so the stated aim in the 2004–2005 Work Plan was to improve gradually [44]. To help with their alignment, the Work Plan also set up action plans focusing on pharmacists, which include reinforcing the pharmaceutical professional qualification system to increase trained licensed pharmacists while keeping the involvement of pharmaceutical technicians in retail pharmacies and increasing the frequency of inspections to help retail pharmacies to align with the new requirements [44]. 

In 2012, the ‘*National Medicine Safety ‘twelve-five’ Programme’* stated that many retail pharmacies still lacked licensed pharmacists and required that any retail pharmacies opened after 2012 be staffed by licensed pharmacists. By the end of 2015, every retail pharmacy had to be owned or managed by a licensed pharmacist and all retail pharmacies had to have licensed pharmacists present during business hours. Any retail pharmacies failing to meet the requirements will have their pharmaceutical trade right withdrawn [50,51]. 

## 3. Discussion

Our review identified four regulations and four strategies that aim to control antibiotic sales in retail pharmacies in China. Regulations and strategies focus on two areas: (i) restricting the dispensing of certain medicines including antibiotics to prescription only, and (ii) ensuring that the dispensing of prescription only medicines is overseen by pharmacists. Central and local FDAs are the main responsible authorities. Initial measures and legal consequences were issued as early as 1999 and enforcement has gradually strengthened over time. Although there has been progress, the aim of restricting the dispensing of antimicrobials in retail pharmacies to prescription-only sales has yet to be achieved [52,53]. One of the key limitations is the availability of qualified staff in retail pharmacies; however, mandated staffing requirements have progressed over time from a pharmaceutical technician as minimum requirement during the early 2000s to a licensed pharmacist being required for new pharmacies opening from 2012, and finally, in all retail pharmacies from 2015. The approach of policies have changed over this time from helping with alignment and gradually improving practices to more strictly regulating practices, with clearer and more serious consequences for violations from reducing business scope to withdrawal of business license. 

Despite the strengthened regulatory framework, the sale of antibiotics by retail pharmacies without a prescription and/or without oversight of a licensed pharmacist remains high in China. Up to 83.6% of retail pharmacies across China sell antibiotics without prescription, although there are regional differences with access easiest in central region and hardest in the eastern region [52,53,54]. National data indicate that the ratio of licensed pharmacists to retail pharmacies increased from 48.1% at the end of 2015 to 70% in March 2017, but there was an unequal distribution of licensed pharmacists among different provinces [55,56]. The proportion of pharmacies having a licensed pharmacist on duty was lower than half in 2017 [52,54]. 

One issue is that the policies themselves lack consistency, clear standards, or necessary mandates. There are eight key pharmaceutical policies issued intermittently over the last two decades including only two specifically focused on antimicrobials. Action plans were very general and lacked follow-up strategies to ensure their implementation, which may explain the minimal change to antimicrobial sales practices in retail pharmacies [14]. Compared with regulations to promote the rational use of antibiotics in hospitals, which specifically set out the roles, responsibilities, and liabilities of all relevant stakeholders (include health administrative authorities, medical institutions, and health care professionals), the measures and penalties to regulate practices of retail pharmacy are less clearly stated [31,35,36]. The penalties for retail pharmacies are much less severe, with only warnings and small financial penalties, whereas violating regulations on the clinical use of antibiotics can contribute to criminal penalties [31]. 

The national policies to control the overuse of antibiotics have been more effective in clinical settings, but may have displaced antibiotic sales from institutions with stringent enforcement to retail pharmacies. Antibiotic sales volumes in clinical settings decreased from 23.80% in 2009 to 19.40% in 2011 [57]. Tighter enforcement of regulations in hospitals and primary health care may result in the only lightly regulated retail pharmacies becoming more significant as a source of antibiotics as consumers may request antibiotics from retail pharmacies when doctors refuse their request for antibiotics [58]. Therefore, policies to reduce excessive use of antibiotics in clinical settings in China may have increased demand in retail pharmacies, pushing excess use into less regulated community settings.

The independent and profit-driven nature of the retail pharmaceutical industry in China makes regulatory interventions more challenging than in the state owned and run hospitals. Unrestricted sales of antibiotics (over the counter, without prescription) have been an important source of income and profit for retail pharmacies [23,59]. Circumvention of the regulations is frequent, with retail pharmacies paying a small amount of money to “rent” a certificate from a licensed pharmacist not actually working in their pharmacy [60], a practice also found in Thailand [61].

China has predominantly relied on guidelines and regulations to improve antibiotic dispensing practices at retail pharmacies, but a complex intervention involving all stakeholders may be more successful. Elsewhere in Asia, multi-component interventions combining regulation enforcement, face-to-face education, and peer support have been more successful at improving antibiotic dispensing practices including asking for prescriptions when selling antibiotics and refusing to sell antibiotics without a prescription [61,62]. An intervention that involved stakeholder groups including health care professionals and the public as well as pharmacists was effective in reducing the OTC antibiotic dispensation rate in community pharmacies from 58.0% in 2010 to 19.1% in 2015 in the Republic of Srpska [63]. 

However, the lack of access or delays in access to antibiotics is as serious an issue as antibiotic resistance, and therefore policies to constrain access can only be implemented effectively alongside policies to ensure appropriate access. The universal provision of antibiotics has been estimated to reduce 75.4% of pneumonia deaths among children under five across 101 countries [64]. Shortages of essential medicines have been identified at multiple levels of the health care system in China and did not improve even after the 2009 health reforms, with over 50% of medicines being reported as out of stock [65,66]. Over-stringent enforcement of regulations that forbid retail pharmacies to sell antibiotics without prescription could reduce access to essential medicines, particularly for people living in rural or poorer areas of China [67,68]. The aim of gradual improvement set out in early policies may indicate governmental recognition of the potential problem with access. There is no single sustainable model in LMICs that increases access while preventing excess; studies exploring the issue of access-excess have suggested context-adjusted programmes, such as applying different restriction models within one country [8,9]. In China, the 2004–2005 Work Plan allowed retail pharmacies in rural areas to follow the local prescribed medicine list to enable the public to access a larger range of prescribed medicines [44], which is a context-based adjustment to ensure access while limiting excess. India similarly has two distribution models where antibiotics distribution is restricted to designated pharmacies and hospitals in urban areas, but is less restricted in rural areas that can be distributed by trained village health workers [9]. This also shares the prescribing task on physicians to community health workers in rural areas where there are few qualified prescribers, which will improve access to appropriate antibiotics. 

Strategies relying on community health workers with brief training and education to increase access to medicines in poor and rural areas, such as Integrated Community Case Management, has been a common practice in LMICs [69], which has been proved to increase the treatment coverage of infectious disease [70,71], thereby limiting excessive antibiotic use by improving appropriate access from trained personnel and depending on the rapid diagnostic tests where possible [72]. Vaccination is another option to reduce the need for antibiotics; a study across 75 countries reported that universal pneumococcal vaccine could reduce 47% of antibiotics used for pneumonia caused by specific strains [8]. Prescription-only regulations would be inadequate and impractical to address the access-excess issue, so a multisectoral intervention needs to take place that considers health services and quality medicines, vaccine and prevention measures, diagnostic technique, guidance and education, and sustainable financing all together.

## 4. Strengths and Limitations

This review focussed on national policy. Since these are all publicly available documents published on official government websites, we are confident that we located all relevant national policies. However, it was not possible to locate local policies, which are often not publicly available. In searching the websites of international organisations and literature databases, the search strategy used the common terms for retail or community pharmacies, and we are confident that we located the most relevant studies. However, like all literature searches, it is possible that we missed studies that did not use the common terms for pharmacies in their title, abstract, or keywords.

## 5. Materials and Methods

### 5.1. Conceptual Framework

We conducted an analysis of policy documents and reports related to controlling antibiotic sales in retail pharmacies in China. The analysis was guided by the concept of the health policy analysis triangle, which was developed for understanding health reforms in LMIC [73]. The authors argue that health policy making is an interactive process where actors and context influence the real-world outcomes from policy changes. This framework identifies the four elements that are key to analysing and understanding health policies and their implementation systematically: political and social-economic contexts (why), the governance processes (how), the actors (who), and the content of policies (what) [73,74] (Figure 2). In this study, the actors included official authorities and retail pharmacies; the context was antibiotic access-excess issues in China; the content was the regulations and strategies set out in new policies; and the process was how these were implemented in practice. 

### 5.2. Literature Search

We used a structured purposive search strategy to search national and international sources for policy documents and reports relevant to antimicrobial stewardship policy and implementation in China’s retail sector [75]. The data sources included the Chinese government and authorities (State Council, National Health Commission, National Medical Products Administration, Certification Centre for Licensed Pharmacist of National Medical Products Administration) and international organisations (World Health Organisation, World Bank, United Nations, and other relevant partners). We also searched English and Chinese Datasets (CNKI and MEDLINE Ovid) for relevant academic studies. Search terms in both English and Mandarin of ‘retail pharmacy’ or ‘community pharmacy’ and ‘antibiotic’ or ‘antimicrobial’ and ‘China’ were used. We then used the snowballing method to include other relevant documents and articles that were mentioned in identified records. We included any policy that would affect retail or community pharmacies and excluded policies that only applied to clinical settings. To ensure the accuracy and completeness of policy content, a list of included key policy documents focusing on antibiotic sales in retail pharmacies was checked with an expert from the authors’ institute. 

### 5.3. Data Analysis

Data were extracted from the records into a matrix in an Excel worksheet with one row per record. Extracted data included year of publication, year of implementation (for regulations only), core content of the policy, and content, processes, context, and actors of policies specifically related to antibiotic sales in retail pharmacies. Data were grouped into categories relating to the four key elements of the health policy analysis triangle and entered in the columns of the matrix. The data were then examined for patterns within and across records and categories. The key policy documents were also organised into a timeframe to examine the development of the policy over time.

## 6. Conclusions

The Chinese government has issued eight key policies since 1999 to control antibiotic sales from retail pharmacies, with measures to restrict the dispensing of certain medicines including antibiotics to prescription only and to ensure that the dispensing of prescribed medicines is overseen by licenced pharmacists. However, the sales of antibiotics without prescription and a lack of licensed pharmacists are still commonly found. This can be related to the lack of consistency of published policies, displacement of demand into less regulated community settings, profit-driven characteristic of retail pharmacies and weak enforcement of regulations, possibly due to the recognition of potential access problems. Multi-faceted interventions developed with the involvement of key stakeholders (local health authorities, retail pharmacy owners, licensed pharmacists, pharmacy staff, and customers) that include consistent follow-up and enforcement strategies (such as regular checks and fines) [14], pharmacy staff and public education, and peer support (where pharmacists working in the same area support each other in regulation adherence) may improve appropriate antibiotic dispensing in retail pharmacies in China. To further address the access-excess issue in China, a system-wide and multisectoral intervention that includes not only prescription-only regulations, but also health services and quality medicines, vaccine and prevention measures, diagnostic techniques, guidance and education, and sustainable financing need to be considered.

## Figures and Tables

**Figure 1 antibiotics-11-00141-f001:**
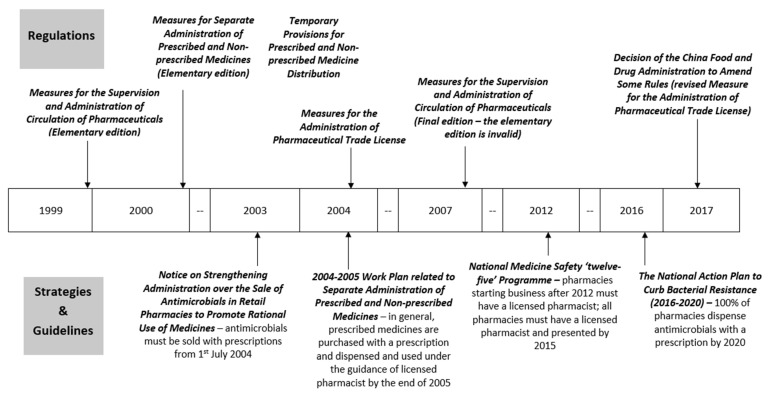
Timeline of key policies related to antibiotic sales in retail pharmacies in China.

**Figure 2 antibiotics-11-00141-f002:**
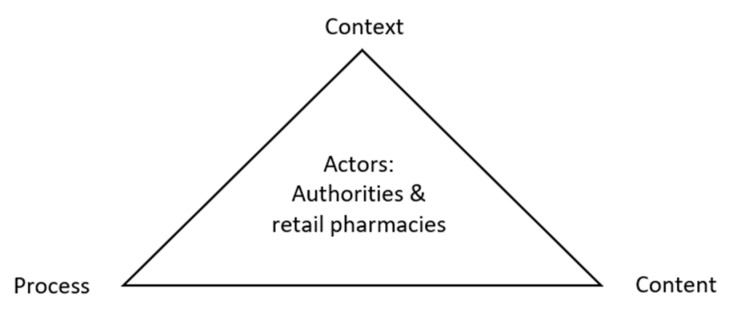
Health policy analysis triangle.

## Data Availability

No new data were created or analysed in this study. Data sharing is not applicable to this article.

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
