# Peer review of "Antibiotic Stewardship in Retail Pharmacies and the Access-Excess Challenge in China: A Policy Review"

_antibiotics, 2022, doi:10.3390/antibiotics11020141_

Round 1
Reviewer 1 Report
The authors present a policy review of the antibiotic stewardship in retail pharmacies in China. The authors discuss a relevant topic, which is the balance between the excess of prescribing antibiotics and the excessive restriction that may lead to uncover patients in deprived areas where Primary Care system is weak according to the authors. They analysed policies during the last decades.
I am very supportive of this study. The manuscript approaches a relevant topic in a extensive, appropriate manner. The manuscript is well written and easy to follow. Therefore, I would suggest publication following minor changes, detailed below:
INTRODUCTION
The introduction covers the state of the art and provide justification for the study. I would only change the last paragraph starting "The aim of this paper was..." instead of "This paper reports on..." in order to clarify that the last paragraph of the introduction covers the objectives, as generally appreciated in scientific papers.
RESULTS
I have nothing to add in this section. I think is complete, clear and properly structured.
DISCUSSION
I would have appreciated some opinion from authors in light of their findings. Do you believe that antibiotic excess-access issue might be improved if Chinese phamarcies accross the country comply with the 2 most problematic issues (give antibiotics only under prescription and having licensed pharmacists)? In which sense? If pharmacies have licensed personnel, should they provide antibiotics withouth prescription in zones where access to primary care is poor?
It is important to provide antibiotics to patients that have no access to health services (especially during the pandemics, when primary care services have been oversaturated and antibiotic prescription has been reduced: 10.1016/j.cmi.2021.02.007, 10.3390/antibiotics10010032). Nevertheless, the increasing number of multi-resistant bacteria, especially carbapenemase producing (10.3390/antibiotics10010089, 10.37201/req/s01.19.2021), prevents us for over-facilitating access to them. It is, therefore, a delicate balance for making decisions at a policy level. Do you have any proposals for China to change according to the current legislation?
Secondly, I would appreciate a paragraph of Limitations of the study in the discussion. It is your search strategy perfect? Do you think that there were no biases or problems in your analyses of the documents?
METHODS
Is this a systematic review? In such case, the philosophy of these documents is to specify how the search was done in order to permit other authors to exactly repeat it. I believe that literature search needs to be further explained in this document. For example, the number of documents identified and discarded in each category (Chinese government and authorities' websites, international organisations' websites, Medline...) should be provided. That would be done through a Flow Chart diagram (you can use PRISMA flow chart style). If possible for authors, please provide these data as a figure.
Again, congratulations for your study, which covers a relevant topic in an appropriate manner.
Author Response
INTRODUCTION
The introduction covers the state of the art and provide justification for the study. I would only change the last paragraph starting "The aim of this paper was..." instead of "This paper reports on..." in order to clarify that the last paragraph of the introduction covers the objectives, as generally appreciated in scientific papers.
Response: Thank you for your suggestion – we have changed the sentence as suggested. Please see line 65-66.
RESULTS
I have nothing to add in this section. I think is complete, clear and properly structured.
DISCUSSION
I would have appreciated some opinion from authors in light of their findings. Do you believe that antibiotic excess-access issue might be improved if Chinese phamarcies accross the country comply with the 2 most problematic issues (give antibiotics only under prescription and having licensed pharmacists)? In which sense? If pharmacies have licensed personnel, should they provide antibiotics withouth prescription in zones where access to primary care is poor?
Response: Our paper shows that policies that just focus on limiting access to antibiotics are not really practicable and may have the unintended consequence of reducing access. We suggest that the policy needs to address the issues of equitable access at the same time as it addresses the issues of restrictions to prevent overuse. We have expanded on this point in by revising and expanding this paragraph in the discussion (line 326-359):
However, the lack of access or delays in access to antibiotics is as serious an issue as antibiotic resistance and therefore policies to constrain access can only be implemented effectively alongside policies to ensure appropriate access. The universal provision of antibiotic has been estimated to reduce 75.4% of pneumonia deaths among children under five across 101 countries [64]. Shortages of essential medicines have been identified at multiple levels of the healthcare system in China and did not improve even after the 2009 health reforms, with over 50% of medicines being reported as out of stock [65, 66]. Over-stringent enforcement of regulations that forbid retail pharmacies to sell antibiotics without prescription could reduce access to essential medicines, particularly for people living in rural or poorer areas of China [67, 68]. The aim of gradual improvement set out in early policies may indicate governmental recognition of the potential problem with access. There has no single sustainable model in LMICs that increases access while preventing excess; studies exploring the issue of access-excess suggested context-adjusted programmes including apply different restriction models within one country [69, 70]. In China, 2004-2005 Work Plan allowed retail pharmacies in rural areas to follow the local prescribed medicine list to enable the public to access a larger range of prescribed medicines [44], which is a context-based adjustment to ensure access while limiting excess. India similarly has two distribution models that antibiotics distribution is restricted to designated pharmacies and hospitals in urban areas but is less restricted in rural areas that can be distributed by trained village health workers [70]. This also shares the prescribing task on physicians to community health worker in rural areas where having few qualified prescribers, which improve access to appropriate antibiotics.
Strategies relying on community health workers with brief training and education to increase access to medicines in poor and rural areas, such as Integrated Community Case Management, has been a common practice in LMICs [71], which has been proved to increase the treatment coverage of infectious disease [72, 73], and thereby limiting excessive antibiotic use by improving appropriate access from trained personnel and depending distribution on the rapid diagnostic tests where possible [74]. Vaccination is another option to reduce the need for antibiotics; a study across 75 countries reported that universal pneumococcal vaccine could reduce 47% of antibiotics used for pneumonia caused by specific strains [69]. Prescription-only regulations would be inadequate and impractical to address the access-excess issue, a multisectoral intervention needs to be taken place that considers health services and quality medicines, vaccine and prevention measures, diagnostic technique, guidance and education and sustainable financing all together.
It is important to provide antibiotics to patients that have no access to health services (especially during the pandemics, when primary care services have been oversaturated and antibiotic prescription has been reduced: 10.1016/j.cmi.2021.02.007, 10.3390/antibiotics10010032). Nevertheless, the increasing number of multi-resistant bacteria, especially carbapenemase producing (10.3390/antibiotics10010089, 10.37201/req/s01.19.2021), prevents us for over-facilitating access to them. It is, therefore, a delicate balance for making decisions at a policy level. Do you have any proposals for China to change according to the current legislation?
Response: We agree completely and we have expanded on this point in the discussion (as shown above) and included examples of possible models from other LMIC facing a similar issue.
Secondly, I would appreciate a paragraph of Limitations of the study in the discussion. It is your search strategy perfect? Do you think that there were no biases or problems in your analyses of the documents?
Response: Thank you and we have added the following paragraph after the discussion section (line 361-369):
Strengths and Limitations
This review focussed on national policy. Since these are all publicly available documents published on official Government websites, we are confident that we located all relevant national policies. However, it was not possible to locate local policies, which are often not publicly available. In searching the websites of international organisations and literature databases, the search strategy used the common terms for retail or community pharmacies and we are confident we located most relevant studies. However, like all literature searches, it is possible that we missed studies that did not use the common terms for pharmacies in their title, abstract or key words.
METHODS
Is this a systematic review? In such case, the philosophy of these documents is to specify how the search was done in order to permit other authors to exactly repeat it. I believe that literature search needs to be further explained in this document. For example, the number of documents identified and discarded in each category (Chinese government and authorities' websites, international organisations' websites, Medline...) should be provided. That would be done through a Flow Chart diagram (you can use PRISMA flow chart style). If possible for authors, please provide these data as a figure.
Response: This is a policy review, not a systematic review, and not all of the elements that would be reported for a systematic review apply. However, we agree that the methods need to be clear enough to allow them to be repeated. We have amended the methods accordingly (line 389-403):
We used a structured purposive search strategy to search national and international sources for policy documents and reports relevant to antimicrobial stewardship policy and implementation in China’s retail sector [77]. The data sources included: Chinese government and authorities (State Council, National Health Commission, National Medical Products Administration, Certification Centre for Licensed Pharmacist of National Medical Products Administration) and international organisations (World Health Organisation, World Bank, Unite Nations and other relevant partners). We also searched English and Chinese Datasets (CNKI and MEDLINE Ovid) for relevant academic studies. Search terms, in both English and Mandarin, of ‘retail pharmacy’ or ‘community pharmacy’ and ‘antibiotic’ or ‘antimicrobial’ and ‘China’ were used. We then used the snowballing method to include other relevant documents and articles that were mentioned in identified records. We included any policy that would affect retail or community pharmacies and excluded policies that only applied to clinical settings. To ensure the accuracy and completeness of policy content, a list of included key policy documents focusing on antibiotic sales in retail pharmacies was checked with an expert from the authors institute.
Again, congratulations for your study, which covers a relevant topic in an appropriate manner
Thank you!

Reviewer 2 Report
This short article reviews the current policies put in place in China towards antibiotic stewardship in community pharmacies. Although this paper has key information that let the reader understand the path that is being built in China towards a safer dispensing of antimicrobials, it fails to contextualise China's reality in the global world. Thus, I would suggest:
1 - Authors could develop more ideas around what other countries similar to China did to overcome the access-excess of antimicrobials in community pharmacies. This discussion would yield key points on how this issue could be addressed. Some ideas were pointed out at the discussion and conclusion, but this should be developed appropriately.
2 - Are there any data that differentiates the tasks/policies given specifically for doctors VS pharmacists VS nurses. In other words, how are other healthcare workers involved in this work?
3 - It would be beneficial to give the authors an idea on how the patients are charged with their prescriptions. Is the Chinese government funding any medication, or are all the medications 100% covered by the patients?
4 - The conclusion is very straightforward and gives a clear idea of what needs to be done. However, ideas need to be expanded. For example, "stakeholders" (line 365) could be designated, "enforcement strategies" which ones, "peer support" what kind of support and given by whom?
Author Response
1 - Authors could develop more ideas around what other countries similar to China did to overcome the access-excess of antimicrobials in community pharmacies. This discussion would yield key points on how this issue could be addressed. Some ideas were pointed out at the discussion and conclusion, but this should be developed appropriately.
Response: Thank you and we agree completely with your suggestions. we have expanded on this point in the discussion (line 326-359) and included examples of possible models from other LMIC facing a similar issue.
However, the lack of access or delays in access to antibiotics is as serious an issue as antibiotic resistance and therefore policies to constrain access can only be implemented effectively alongside policies to ensure appropriate access. The universal provision of antibiotic has been estimated to reduce 75.4% of pneumonia deaths among children under five across 101 countries [64]. Shortages of essential medicines have been identified at multiple levels of the healthcare system in China and did not improve even after the 2009 health reforms, with over 50% of medicines being reported as out of stock [65, 66]. Over-stringent enforcement of regulations that forbid retail pharmacies to sell antibiotics without prescription could reduce access to essential medicines, particularly for people living in rural or poorer areas of China [67, 68]. The aim of gradual improvement set out in early policies may indicate governmental recognition of the potential problem with access. There has no single sustainable model in LMICs that increases access while preventing excess; studies exploring the issue of access-excess suggested context-adjusted programmes including apply different restriction models within one country [69, 70]. In China, 2004-2005 Work Plan allowed retail pharmacies in rural areas to follow the local prescribed medicine list to enable the public to access a larger range of prescribed medicines [44], which is a context-based adjustment to ensure access while limiting excess. India similarly has two distribution models that antibiotics distribution is restricted to designated pharmacies and hospitals in urban areas but is less restricted in rural areas that can be distributed by trained village health workers [70]. This also shares the prescribing task on physicians to community health worker in rural areas where having few qualified prescribers, which improve access to appropriate antibiotics.
Strategies relying on community health workers with brief training and education to increase access to medicines in poor and rural areas, such as Integrated Community Case Management, has been a common practice in LMICs [71], which has been proved to increase the treatment coverage of infectious disease [72, 73], and thereby limiting excessive antibiotic use by improving appropriate access from trained personnel and depending distribution on the rapid diagnostic tests where possible [74]. Vaccination is another option to reduce the need for antibiotics; a study across 75 countries reported that universal pneumococcal vaccine could reduce 47% of antibiotics used for pneumonia caused by specific strains [69]. Prescription-only regulations would be inadequate and impractical to address the access-excess issue, a multisectoral intervention needs to be taken place that considers health services and quality medicines, vaccine and prevention measures, diagnostic technique, guidance and education and sustainable financing all together.
2 - Are there any data that differentiates the tasks/policies given specifically for doctors VS pharmacists VS nurses. In other words, how are other healthcare workers involved in this work?
Response: This review focussed exclusively on policies that affect retail pharmacies. No doctors, nurses or other health care workers are involved in retail pharmacies. There are policies that are specific to doctors and pharmacists working in clinical setting (we have clarified this in the context in line 104-106) but this was not the subject of this review.
Coupled with these reforms, a new strategy, National Special Campaign, ran from 2011 to 2013 and centred on the regulations for the clinical use of antibiotics. These policies focused on the clinical settings with clear tasks and responsibilities related to the appropriate use of antibiotics set up for physicians, pharmacists, microbiologists, and administrators in healthcare institutions [31-34].
3 - It would be beneficial to give the authors an idea on how the patients are charged with their prescriptions. Is the Chinese government funding any medication, or are all the medications 100% covered by the patients?
Response: Information about payment mechanisms, which is generally through one of the main health insurance systems, has been added to the introduction (line 53-60).
The cost of antibiotics can be covered by health insurance. China’s health insurance system is based around three basic medical insurance schemes and covered over 95% of the population, including Urban Employee Basic Medical Insurance (UEBMI) and Urban Resident Basic Medical Insurance (URBMI) for urban areas and New Rural Cooperative Medical Scheme (NRCMS) for rural areas. Patients covered by UEBMI can use their medical savings account to pay outpatient medical services and medications, while URBMI and NRCMS will cover approximately 50% (varied in regions) of outpatient medical services and medications for patients joined these schemes [17].
4 - The conclusion is very straightforward and gives a clear idea of what needs to be done. However, ideas need to be expanded. For example, "stakeholders" (line 365) could be designated, "enforcement strategies" which ones, "peer support" what kind of support and given by whom?
Response: Thank you for your suggestions. We have amended the conclusion to provide additional detail (line 423-433):
Multi-faceted interventions developed with the involvement of key stakeholders (local health authorities, retail pharmacy owners, licensed pharmacists, pharmacy staff & customers) that include consistent follow-up and enforcement strategies (such as regular checks and fines) [14], pharmacy staff and public education and peer support (where pharmacists working in the same area support each other in regulation adherence) may improve appropriate antibiotic dispensing in retail pharmacies in China. To further address the access-excess issue in China, a system-wide and multisectoral intervention includes not only prescription-only regulations, but also health services and quality medicines, vaccine and prevention measures, diagnostic technique, guidance and education and sustainable need to be considered.

Round 2
Reviewer 2 Report
Dear authors, my replies are below, in red:
1 - Authors could develop more ideas around what other countries similar to China did to overcome the access-excess of antimicrobials in community pharmacies. This discussion would yield key points on how this issue could be addressed. Some ideas were pointed out at the discussion and conclusion, but this should be developed appropriately.
Response: Thank you and we agree completely with your suggestions. we have expanded on this point in the discussion (line 326-359) and included examples of possible models from other LMIC facing a similar issue.
However, the lack of access or delays in access to antibiotics is as serious an issue as antibiotic resistance and therefore policies to constrain access can only be implemented effectively alongside policies to ensure appropriate access. The universal provision of antibiotic has been estimated to reduce 75.4% of pneumonia deaths among children under five across 101 countries [64]. Shortages of essential medicines have been identified at multiple levels of the healthcare system in China and did not improve even after the 2009 health reforms, with over 50% of medicines being reported as out of stock [65, 66]. Over-stringent enforcement of regulations that forbid retail pharmacies to sell antibiotics without prescription could reduce access to essential medicines, particularly for people living in rural or poorer areas of China [67, 68]. The aim of gradual improvement set out in early policies may indicate governmental recognition of the potential problem with access. There has no single sustainable model in LMICs that increases access while preventing excess; studies exploring the issue of access-excess suggested context-adjusted programmes including apply different restriction models within one country [69, 70]. In China, 2004-2005 Work Plan allowed retail pharmacies in rural areas to follow the local prescribed medicine list to enable the public to access a larger range of prescribed medicines [44], which is a context-based adjustment to ensure access while limiting excess. India similarly has two distribution models that antibiotics distribution is restricted to designated pharmacies and hospitals in urban areas but is less restricted in rural areas that can be distributed by trained village health workers [70]. This also shares the prescribing task on physicians to community health worker in rural areas where having few qualified prescribers, which improve access to appropriate antibiotics.
Strategies relying on community health workers with brief training and education to increase access to medicines in poor and rural areas, such as Integrated Community Case Management, has been a common practice in LMICs [71], which has been proved to increase the treatment coverage of infectious disease [72, 73], and thereby limiting excessive antibiotic use by improving appropriate access from trained personnel and depending distribution on the rapid diagnostic tests where possible [74]. Vaccination is another option to reduce the need for antibiotics; a study across 75 countries reported that universal pneumococcal vaccine could reduce 47% of antibiotics used for pneumonia caused by specific strains [69]. Prescription-only regulations would be inadequate and impractical to address the access-excess issue, a multisectoral intervention needs to be taken place that considers health services and quality medicines, vaccine and prevention measures, diagnostic technique, guidance and education and sustainable financing all together.
This fully delivers the suggestions made. Very good improvement.
2 - Are there any data that differentiates the tasks/policies given specifically for doctors VS pharmacists VS nurses. In other words, how are other healthcare workers involved in this work?
Response: This review focussed exclusively on policies that affect retail pharmacies. No doctors, nurses or other health care workers are involved in retail pharmacies. There are policies that are specific to doctors and pharmacists working in clinical setting (we have clarified this in the context in line 104-106) but this was not the subject of this review.
Coupled with these reforms, a new strategy, National Special Campaign, ran from 2011 to 2013 and centred on the regulations for the clinical use of antibiotics. These policies focused on the clinical settings with clear tasks and responsibilities related to the appropriate use of antibiotics set up for physicians, pharmacists, microbiologists, and administrators in healthcare institutions [31-34].
Thank you for your reply. This fully explains the scope of the paper. No additional changes are needed.
3 - It would be beneficial to give the authors an idea on how the patients are charged with their prescriptions. Is the Chinese government funding any medication, or are all the medications 100% covered by the patients?
Response: Information about payment mechanisms, which is generally through one of the main health insurance systems, has been added to the introduction (line 53-60).
The cost of antibiotics can be covered by health insurance. China’s health insurance system is based around three basic medical insurance schemes and covered over 95% of the population, including Urban Employee Basic Medical Insurance (UEBMI) and Urban Resident Basic Medical Insurance (URBMI) for urban areas and New Rural Cooperative Medical Scheme (NRCMS) for rural areas. Patients covered by UEBMI can use their medical savings account to pay outpatient medical services and medications, while URBMI and NRCMS will cover approximately 50% (varied in regions) of outpatient medical services and medications for patients joined these schemes [17].
Many thanks for your answer. This will really improve the paper and make it clearer for healthcare professionals outside China.
4 - The conclusion is very straightforward and gives a clear idea of what needs to be done. However, ideas need to be expanded. For example, "stakeholders" (line 365) could be designated, "enforcement strategies" which ones, "peer support" what kind of support and given by whom?
Response: Thank you for your suggestions. We have amended the conclusion to provide additional detail (line 423-433):
Multi-faceted interventions developed with the involvement of key stakeholders (local health authorities, retail pharmacy owners, licensed pharmacists, pharmacy staff & customers) that include consistent follow-up and enforcement strategies (such as regular checks and fines) [14], pharmacy staff and public education and peer support (where pharmacists working in the same area support each other in regulation adherence) may improve appropriate antibiotic dispensing in retail pharmacies in China. To further address the access-excess issue in China, a system-wide and multisectoral intervention includes not only prescription-only regulations, but also health services and quality medicines, vaccine and prevention measures, diagnostic technique, guidance and education and sustainable need to be considered.
This is a very good improvement. Conclusion has been completed with more information, so that no answers are left in blank.
All suggestions were appropriately discussed and addressed. I fully endorse the publishing of this article.